# Aligning staff schedules, testing, and isolation reduces the risk of COVID-19 outbreaks in carceral and other congregate settings: A simulation study

Christopher M. Hoover[1]*, Nicholas K. Skaff[2], Seth Blumberg[1,3‡], Rena Fukunaga[2‡]

1 Francis I. Proctor Foundation, University of California, San Francisco, San Francisco, California, United States of America, 2 Centers for Disease Control and Prevention, Atlanta, Georgia, United States of America, 3 Division of Hospital Medicine, Department of Medicine, University of California, San Francisco, San Francisco, California, United States of America

‡ SB and RF jointly supervised this work.
* cmhoove14@gmail.com

**Data Availability Statement:** Anonymized staff records used to generate figure 2 and inform staff schedules used in simulations can be requested by

## Abstract

COVID-19 outbreaks in congregate settings remain a serious threat to the health of disproportionately affected populations such as people experiencing incarceration or homelessness, the elderly, and essential workers. An individual-based model accounting for individual infectiousness over time, staff work schedules, and testing and isolation schedules was developed to simulate community transmission of SARS-CoV-2 to staff in a congregate facility and subsequent transmission within the facility that could cause an outbreak. Systematic testing strategies in which staff are tested on the first day of their workweek were found to prevent up to 16% more infections than testing strategies unrelated to staff schedules. Testing staff at the beginning of their workweek, implementing timely isolation following testing, limiting test turnaround time, and increasing test frequency in high transmission scenarios can supplement additional mitigation measures to aid outbreak prevention in congregate settings.

## Introduction

Throughout the COVID-19 pandemic, outbreaks in congregate settings such as skilled nursing facilities [1], homeless shelters [2–5], and carceral (e.g., prisons and jails) facilities [6] have been devastating. Staff have inadvertently introduced SARS-CoV-2, the virus that causes COVID-19, into congregate settings [6–8], making routine testing of staff and subsequent isolation of infectious staff an essential strategy to mitigate case importation to resident populations. Prior analyses suggest that routine SARS-CoV-2 screening testing can reduce transmission in homeless shelters [9], in healthcare settings [10], and during airline travel [11].

In correctional and detention facilities, preventing spillover from the community to facility staff and subsequently into resident populations remains one of many challenges to limit

submitting a Data Request Form for consideration by the California Correctional Health Care Services (CCHCS) Data Advisory Committee (DAC). All simulations, analyses, and visualizations were compiled in R software version 4.0.4 with aid from the tidyverse, triangle, and patchwork packages. Code is available at https://github.com/cmhoove14/Congregate-Staff-Testing.

**Funding:** CMH and SB were supported by CDC U01CK000590 awarded to SB, as part of the Modeling Infectious Diseases in Healthcare (MInD) Network. Funders played no role in study design, data collection or analysis, preparation of the manuscript, or decision to publish.

**Competing interests:** The authors have declared that no competing interests exist. The findings and conclusions in this report are those of the author(s) and do not necessarily represent the official position of the U.S. Department of Health and Human Services, the Centers for Disease Control and Prevention, or the authors' affiliated institutions.

SARS-CoV-2 transmission [12]. Due to the high risk of SARS-CoV-2 transmission in congregate settings [6] and the emergence of SARS-CoV2 variants with substantial ability to evade prior immunity, questions remain around optimal testing policies for staff, regardless of vaccination status, with reports of infections in vaccinated persons in large public gatherings [13], as well as in congregate settings such as health care [14], and correctional [15] facilities.

At this time, the CDC Guidance on Prevention and Management of Coronavirus Disease 2019 (COVID-19) in Correctional and Detention Facilities [16] does not specify when staff should be tested during the workweek to minimize the spread of SARS-CoV-2 via rapid identification and isolation of new staff cases. The timing of systematic testing in relation to work schedules and variable infectiousness profiles could have profound importance for designing optimal systematic testing strategies and for informing downstream activities to prevent transmission, such as isolation of positive staff cases. Testing early in the work week may miss recently acquired infections and lead to staff working around the time of their peak infectiousness. However, testing later in the work week risks missing infectious individuals who are allowed to work prior to being isolated.

This study examines the relationship between work schedules, testing schedules, and within-facility transmission. An analytic framework to estimate the effect of variable testing frequencies and turnaround time between test administration and isolation on SARS-CoV-2 transmission is presented. In addition, an individual-based model which incorporates work and testing schedules influenced by those observed in operations records collected by the California Department of Corrections and Rehabilitation (CDCR) is used to simulate community acquisition of SARS-CoV-2 by staff and subsequent transmission in a congregate setting. Simulations exploring the impact of aligning testing schedules with work schedules are conducted across testing frequency, background community infection rate, and within-facility transmission rate.

## Methods

### Model framework and parameterization for SARS-CoV-2

Building on previous work investigating the effects of non-pharmaceutical interventions [17] and testing [18] on the transmission of infectious diseases, individual contributions to SARS-CoV-2 transmission through time were modeled from an infectiousness profile, $\beta_t$, here derived from the probability density function of the triangle distribution, with infectiousness beginning after the latent period, ending after the duration of the infectious period, and peaking at some point in between ($a = t_{latent}$, $b = t_{total}$ where $t_{total} = t_{infectious} + t_{latent}$, $c = t_{peak}$, and a<c<b; Fig 1A).

The viral dynamics of SARS-CoV-2 make control efforts challenging, as high infectiousness in the absence of symptoms is common [19–21]. For SARS-CoV-2, peak infectiousness ($t_{peak}$) tends to coincide with the onset of symptoms (for cases that are symptomatic), but occurs after completion of the latent period (i.e. $t_{peak} \approx t_{incubation}$ and $t_{incubation} > t_{latent}$) [21]. The expected number of new cases generated by an individual at time $t$ is thus $r_t = R\beta_t$, where $R$ is the effective reproduction number, here defined as the expected number of cases generated in a facility by a new case over the duration of their infectious period, assuming they spent their entire infectious period in the facility. Table 1 lists the distributions of $t_{incubation}$, $t_{latent}$, and $t_{infectious}$ used here.

Isolating infectious individuals prior to the end of their infectious period, $t_{total}$, through contact tracing, self-isolation following the onset of symptoms, or isolation following a positive test result, reduces $R$ according to: $R_{iso} = R(1 - \int_{t_{iso}}^{t_{total}} \beta_t dt)$, where $t_{iso}$ is the time at which isolation occurs. Reducing $R_{iso}$ can thus be represented as removing a larger slice from the overall

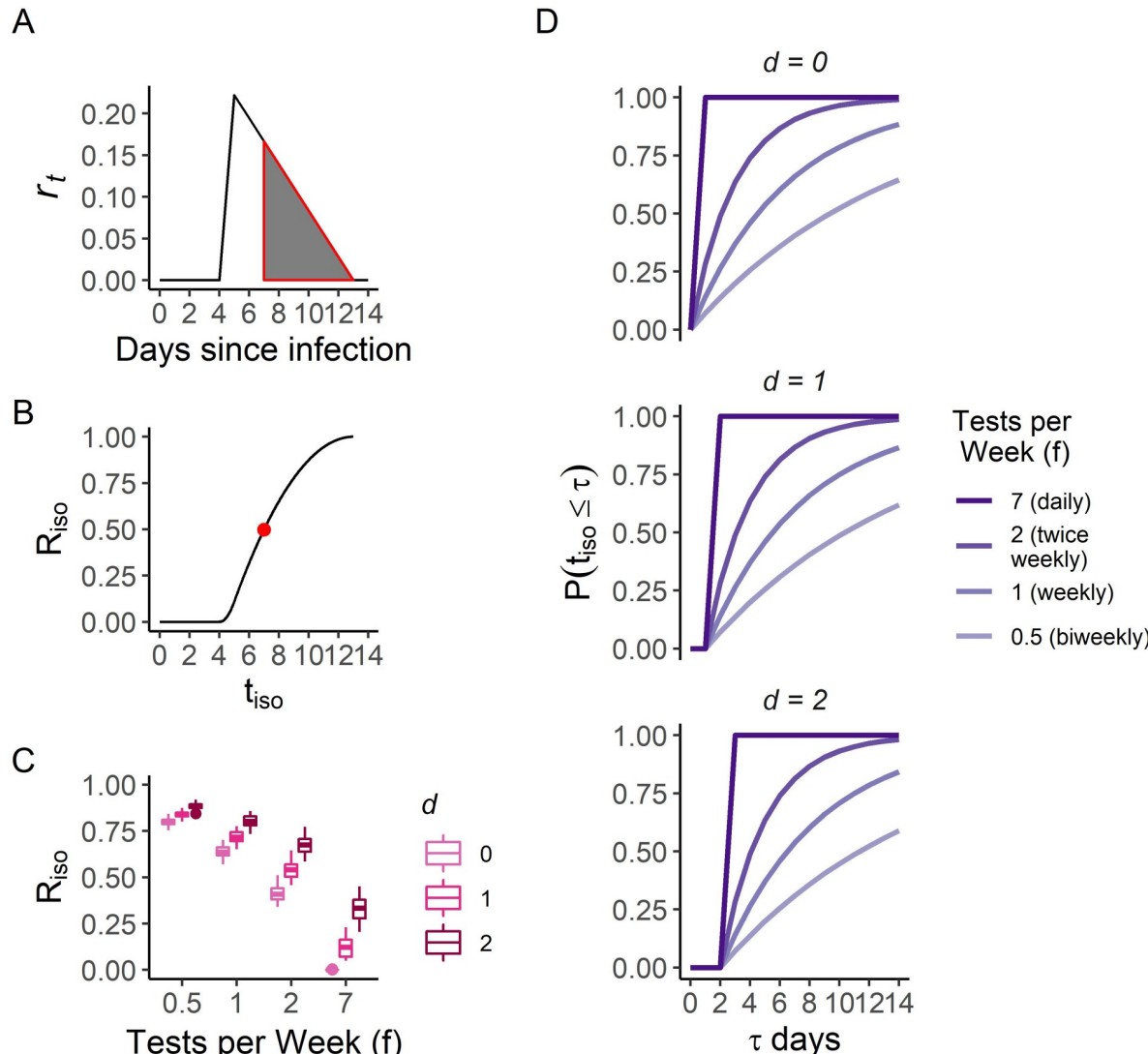

**Fig 1. Analytic framework exploring effects of variable infectiousness through time, testing frequencies, and delays on SARS-CoV-2 transmission.** A) Example infectiousness profile for $R = 1$, $t_{latent} = 4$, $t_{incubation} = 5$, $t_{infectious} = 9$, with line indicating infectiousness ($r_t$) through time and shaded area demonstrating infectiousness slice removed if $t_{iso} = 7$, leading to $R_{iso} = 0.50$. B) $R_{iso}$ as a function of $t_{iso}$ with same parameters as in A and point indicating scenario depicted in A. C) Boxplots showing distributions of $R_{iso}$ as a function of testing frequency, $f$, and delay in obtaining test results, $d$, incorporating uncertainty in $t_{latent}$, $t_{incubation}$, and $t_{infectious}$ by drawing $n = 100$ parameter sets for each, with baseline $R = 1$. Boxplots indicate median, interquartile range, and full range of values of $R_{iso}$. D) Probability isolation occurs as a function of testing frequency, $f$, delay in obtaining test results, $d$, and days from exposure to isolation $\tau$, i.e. $t_{iso} \leq \tau$, demonstrating that delays in obtaining test results substantially reduce the probability of prompt isolation, particularly among most frequent testing scenarios.

infectiousness triangle by reducing $t_{iso}$ (Fig 1A). Fig 1B shows the relationship between $R_{iso}$ and $t_{iso}$ is sigmoidal, implying the benefits of isolation level off later in the infectious period.

Next, to explore the influence of testing on $t_{iso}$ and $R_{iso}$, test frequency, $f$, is defined as the average number of tests per week. Assuming testing is done randomly through time and is independent of symptoms or known contacts, the probability of going $t$ days without being tested can be estimated as $(1-f/7)^t$, where, for example $f = 1$ if testing is conducted weekly. The probability that isolation has occurred by day $\tau$ after onset of infectiousness can be estimated as: $P(t_{iso} \leq \tau) = 1 - (1-f/7)^\tau$ if isolation occurs immediately after testing. Given substantial

**Table 1. Distributions and parameter values used in analytic framework and model simulations.** The latent period is defined as the time between exposure and onset of infectiousness, the incubation period as the time between exposure and both symptoms and peak infectiousness (even in the absence of symptoms), and the infectious period as the total time a case is infectious.

| Parameter | Distribution | Source |
|---|---|---|
| Incubation Period ($t_{incubation}$) | *Lognormal* (1.63, 0.5) | [22] |
| Latent Period ($t_{latent}$) | $t_{incubation}$−*Uniform* (0, 2) | [21, 23] |
| Infectious Period ($t_{infectious}$) | *Uniform* (7, 10) | [21, 23] |

turnaround times between testing and isolation, particularly when using nucleic acid amplification tests (NAATs), the delay, $d$, between testing and isolation can be incorporated as: $P(t_{iso} = \tau) = 0$ for $\tau < d$ and $P(t_{iso} = \tau) = 1 - (1 - f/7)^{\tau - d}$ for $\tau \geq d$. Fig 1D shows that delays have a detrimental effect on prompt isolation, particularly by making isolation prior to the delay ($t_{iso} < d$) impossible.

Testing frequency and isolation delays can also be incorporated into estimation of $R_{iso}$, with the reduction in $R$ due to isolation estimated from infectiousness on day $t$ weighted by the probability of being isolated on day $t$. Discretizing, this gives:

$$R_{iso} = R - \sum_{t=t_{latent}+d}^{t_{tot}} r_t \left( 1 - \left( 1 - \frac{f}{7} \right)^{t - t_{latent} - d} \right)$$

Fig 1C shows distributions of $R_{iso}$ that incorporate uncertainty in the SARS-CoV-2 latent, incubation, and total infectious periods, across test frequencies ranging from daily ($f = 7$) to biweekly ($f = 0.5$) and isolation delays from 0 to 2 days. $R_{iso}$ is similar when testing every day ($f = 7$) with a two-day isolation delay ($d = 2$) vs testing twice per week ($f = 2$) with immediate isolation ($d = 0$) (Fig 1C, median $\mathcal{R}_{iso}(d = 0, f = 2) = 0.42$ and $\mathcal{R}_{iso}(d = 2, f = 7) = 0.33$, respectively), reiterating the importance of reducing delays between testing and isolation.

## Individual-based model simulations

Model setup. An individual-based model (IBM) building on the framework described above and incorporating staff working and testing schedules was developed to simulate SARS-CoV-2 transmission within a congregate facility. The main priority of the IBM was to investigate how testing and staffing schedules should be configured to optimally prevent transmission in a congregate facility.

Staff move through susceptible (S), exposed (E), infectious (I), and recovered (R) states after acquiring infection from the community according to the community prevalence when they are not working, or from fellow staff while working, according to the force of infection described below. Parameters for newly exposed staff are drawn to determine $t_{latent}$, $t_{incubation}$, and $t_{infectious}$, from which an infectiousness profile, $\beta_{it}$ is generated. Tested staff produce a positive test result if $\beta_{it} > 0$ [24–26] and no testing other than systematic screening testing occurs. Positive staff enter an isolated (O) state immediately if $d = 0$. If there is a delay between testing and isolation ($d > 0$), staff first enter a tested (T) state before O, during which time they may continue to work while infectious. Recovered staff are assumed to remain in state R and not return to state S due to the short time frame of the simulation. Staff in state O are restricted from working for 10 days and are not required to undergo screening testing for 90 days following a positive result.

The simulated number of infections caused by staff is estimated from each simulation from the total time spent in the facility while infectious, weighted by infectiousness:

$I_{sim}^{tot} = \sum_{t=1}^{t_{sim}} \sum_{i=1}^{w} R\beta_{it} W(w_{it})$, where $t_{sim}$ = 540 work shifts (3 shifts per day for 180 days), $w$ = 700 staff in the facility (derived from the average of all CDCR facilities), and $W(w_{it})$ is an indicator function derived from individual staff work schedules (described below) that defines when each worker is in the facility. The related quantity $\lambda_{it}^{work} = \frac{\sum_{i=1}^{w} R\beta_{it} W(w_{it})}{\sum_{i=1}^{w} W(w_{it})}$ is the force of infection for staff who are at work at time $t$.

Staffing and testing strategies. Information on workdays (e.g., Mon-Thurs), work shifts (e.g., morning, evening, night), and SARS-CoV-2 testing schedules for custody staff were derived from CDCR operations records and used to generate staff working schedules in model simulations. Standard work schedules including typical workdays during the week and typical shift worked were identified using K-means clustering. The most common work schedules identified among CDCR staff were then used directly to generate work schedules in the IBM.

Two experimental testing strategies were considered in model simulations. Under a random testing strategy, testing for each worker occurs at random during their work shifts depending on the frequency (i.e. with $f$ = 2, workers would be tested during two of their shifts, chosen at random each workweek). Under a systematic testing strategy, each worker is always tested on the same day(s) each week. For $f$ = 1, systematic testing always occurs on the first day of their workweek; for $f$ = 2, systematic testing always occurs on the first and third days; and for $f$ = 4, testing occurs on each of the first four workdays in a workweek.

The total number of tests conducted in each simulation is recorded as: $\mathcal{T}_{sim}^{tot} = \sum_{t=1}^{t_{sim}} \sum_{i=1}^{w} \mathcal{T}(w_{it})$, where $T(w_{it})$ is an indicator function defining shifts at which staff are tested based on their testing schedule. Combined with the expected number of cases in the simulation, the incremental test effectiveness ratio (ITER) is estimated as: $\text{ITER} = \frac{\mathcal{T}_{sim}^{tot}}{\mathcal{I}_{sim}^{tot} - \mathcal{I}_{ref}^{tot}}$, where $\mathcal{I}_{ref}^{tot}$ is the number of infections in a reference scenario with no testing. The ITER can be interpreted as the number of tests needed to prevent one infection in the simulation scenario being evaluated.

Sensitivity analyses. Simulations were conducted across community prevalences of 0.1%, 0.5%, and 1%, corresponding to values found in [27]; $\mathcal{R} \in \{0.5, 1.0, 1.5\}$; $d \in \{0, 1\}$; and $f \in \{0.5, 1, 2, 4\}$. Sensitivity analyses investigating the influence of self-isolation of symptomatic infections and imperfect test sensitivity are explained in the supplement.

Code and data availability. Anonymized staff records used to generate Fig 2 and inform staff schedules used in simulations can be requested by submitting a Data Request Form for consideration by the California Correctional Health Care Services (CCHCS) Data Advisory Committee (DAC). All simulations, analyses, and visualizations were compiled in R software version 4.0.4 [28] with aid from the tidyverse [29], triangle [30], and patchwork [31] packages. Code is available at https://github.com/cmhoove14/Congregate-Staff-Testing.

## Human subjects

This study is exempt from IRB review because it involved no identifiable data and no patient enrollment.

## Results

### Staff working and testing schedules

There were 4,248,692 staff workdays consisting of the date and shift (morning, evening, or night) in CDCR operations records collected from March 1, 2020 to February 28, 2021. Of these, 2,849,801(67.1%) occurred as part of workweeks with at least 4 consecutive workdays.

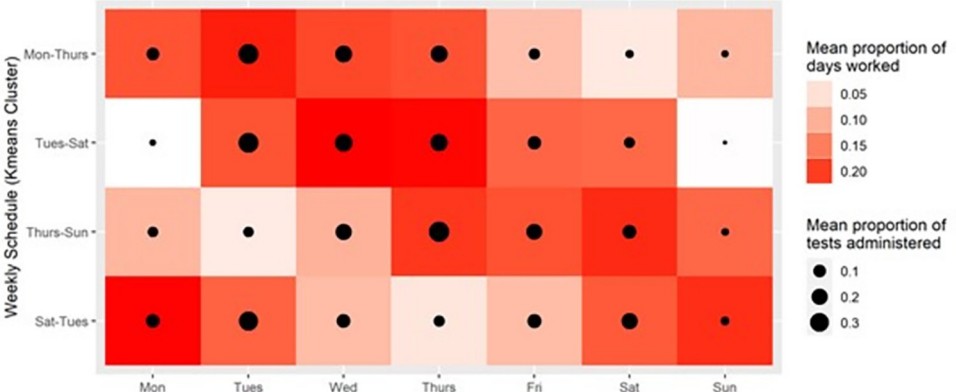

**Fig 2. Staff work and testing schedules.** Four typical weekly work schedules (y-axis) were identified among CDCR custody staff. These include a Monday to Thursday workweek (21% of staff), a Tuesday to Saturday workweek (33% of staff), a Thursday to Sunday workweek (22% of staff), and a Saturday to Tuesday workweek (24% of staff). The red shading shows the mean proportion of staff workdays that consist of a particular day of the week (x-axis; i.e. darker shades of red indicate that staff with the specified schedule more commonly worked on that day). The size of the black circles represents the mean proportion of the total number of tests administered to each group that were given on the specified day.

The first day of the workweek varied across staff, though four typical workweek schedules starting on Monday, Tuesday, Thursday, or Saturday and continuing for the next three days were identified using K-means clustering (Fig 2). Staff adding a variable fifth day to their work-week was common, though this fifth day was less predictable (Fig A in S1 File). Staff typically worked eight hours during either the morning, evening, or night shift, though alternating between morning and evening shifts, and taking on an additional shift was also common. These same work schedules were used to generate staff schedules in model simulations.

Tests were most often administered on Tuesdays (if the staff had Tuesday in their typical workweek) regardless of whether it was the first day of the staff's workweek (Fig 2). Testing on Wednesday and Thursday was also common across work schedules. 46.3% of test results were returned on the same day as specimen collection, 44.7% on the day after specimen collection, and 8.9% two or more days after specimen collection. Of 467,370 total SARS-CoV-2 staff tests, 89,617 (19.2%) were administered on the first day of a workweek consisting of 4 or more consecutive workdays, corresponding to a systematic testing strategy with $f = 1$ described above.

## Simulation results

Systematic testing strategies prevented more infections in simulated facilities than random testing strategies. Fig 3 shows a comparison of the number of infections generated ($\mathcal{I}_{sim}^{tot}$) when implementing each testing strategy across testing frequencies, levels of community prevalence, and within-facility $\mathcal{R}$ with either no delay or a one-day isolation delay. In the highest transmission scenario ($CP = 1\oslash, \mathcal{R} = 1.5$), no testing led to a median $\mathcal{I}_{sim}^{tot} = 111.65$ (IQR 108.13–114.73) expected infections. Testing randomly once per week with no isolation delay resulted in a median $\mathcal{I}_{sim}^{tot} = 47.48$ (IQR 44.52–50.3; Fig 3 right panel, rightmost yellow circle), whereas testing systematically on the first day of the work week with no isolation delay resulted in $\mathcal{I}_{sim}^{tot} = 31.31$ (IQR 29.48–33.21; Fig 3 right panel, rightmost yellow square). However, systematic testing accompanied by a one day isolation delay leads to $\mathcal{I}_{sim}^{tot} = 53.71$ (IQR 50.98–55.87; Fig 3 right panel, rightmost yellow cross).

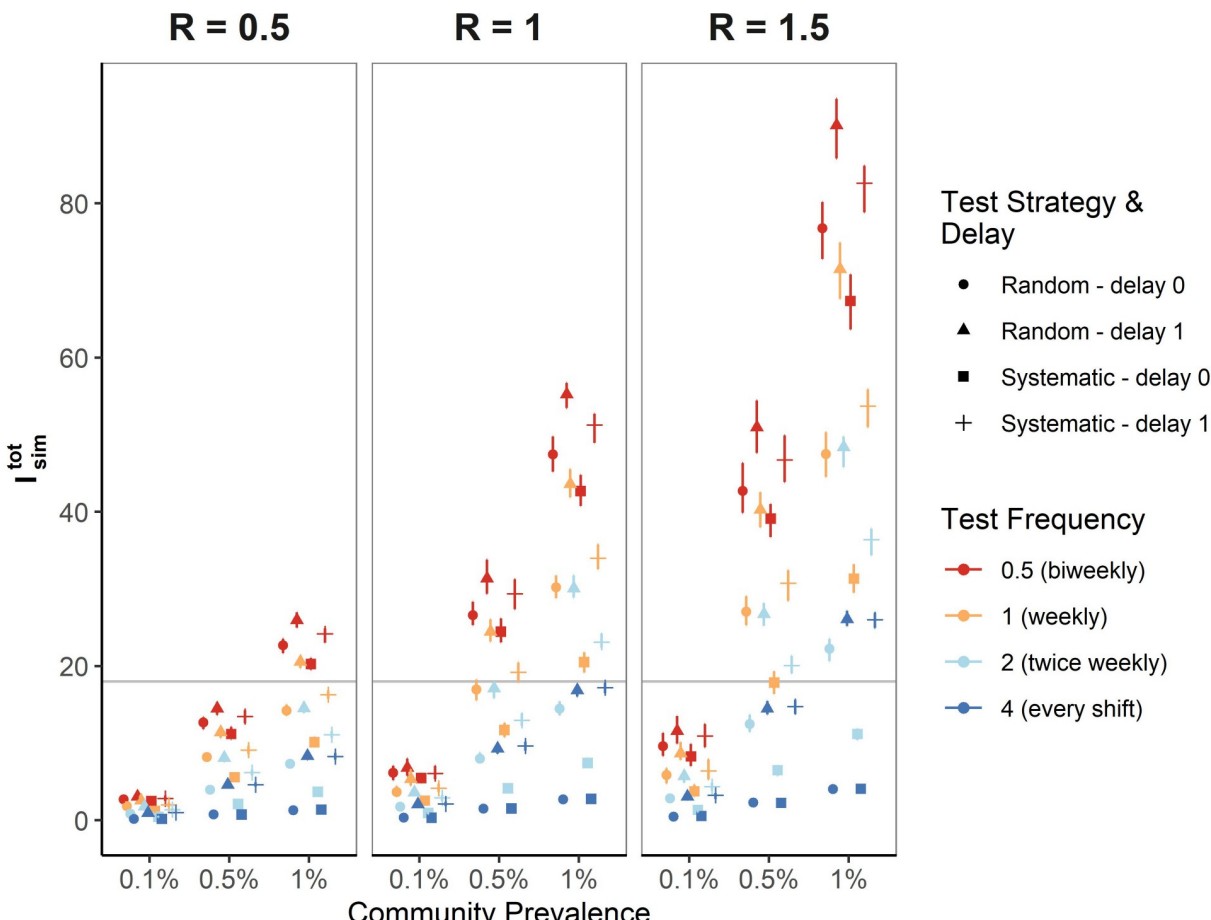

**Fig 3. Number of expected infections generated in a facility from model simulations comparing random and systematic testing strategies across transmission scenarios, test frequencies, and delays isolating infectious individuals who have tested positive.** Systematic testing strategies (■, ✚) prevent more infections than random strategies (●, ▲) across all transmission scenarios (indicated by community prevalence across the x axis and by reproduction number across the panels) and test frequencies (indicated by different colored symbols with blue corresponding to the highest test frequency of 4 tests per week and red the lowest test frequency of biweekly testing). More infections are expected in transmission scenarios with higher within-facility $\mathcal{R}$ and higher community prevalence. Preventing delays between testing and isolation of positives (squares compared to crosses and triangles compared to circles) and increasing test frequency (red = lowest frequency, blue = highest frequency) also reduces the number of infections. The horizontal gray line serves as a reference to assess the testing frequency needed to maintain $\mathcal{I}_{sim}^{tot} \leq 18$ (corresponding to one infection every ten days) across different transmission scenarios. Error bars represent the interquartile range of $\mathcal{I}_{sim}^{tot}$ derived from 100 simulations per scenario run for 180 days among 700 staff.

The horizontal gray line in Fig 3 demonstrates a potential threshold number of infections to avoid exceeding at $\mathcal{I}_{sim}^{tot} = 18.00$, corresponding to an average of one infection within the simulated facility every ten days. Implementing a systematic rather than random–testing strategy can be sufficient to prevent $\mathcal{I}_{sim}^{tot}$ from exceeding such a threshold without changing the frequency in many transmission scenarios (e.g. compare circles to squares of the same color in Fig 3) though in the highest transmission scenarios, greater than twice-weekly testing may be needed. Table 2 additionally shows the testing frequency in tests per week under a systematic testing strategy necessary to ensure that the upper quartile of expected infections is maintained below this threshold.

Across all transmission scenarios, biweekly systematic testing with no isolation delay averted an average of 40% of transmissions that would have occurred with no testing, while random testing averted an average of 33% of transmissions. For weekly frequency, systematic

**Table 2. Test frequency to prevent excessive infections.** Test frequency (tests per week) under a systematic testing strategy needed to maintain the upper quartile of expected infections in the simulated facility below a threshold of 1 every ten days across transmission scenarios conveyed by the within-facility basic reproduction number ($\mathcal{R}$), community prevalence (CP), and delays between testing and isolation of infectious workers.

|  | $R = 0.5$ | $R = 1$ | $R = 1.5$ |
|---|---|---|---|
| *Delay = 0* |  |  |  |
| CP = 0.1% | 0 | 0 | 0 |
| CP = 0.5% | 0.5 | 1 | 2 |
| CP = 1% | 1 | 2 | 2 |
| *Delay = 1* |  |  |  |
| CP = 0.1% | 0 | 0 | 0.5 |
| CP = 0.5% | 0.5 | 2 | 4 |
| CP = 1% | 1 | 4 | 4+ |

testing averted an average of 71% of transmissions versus 57% of transmissions when testing randomly; and for twice weekly testing, systematic testing averted an average of 90% of transmissions versus 80% of transmissions when testing randomly.

The ITER, interpreted as the number of tests needed per infection averted due to testing, is also useful to aid decision making, particularly in resource-constrained settings. Fig 4 shows estimates of the ITER across transmission scenarios, test strategies, and test frequencies. In the highest transmission scenario ($\mathcal{R} = 1.5$, 1% community prevalence), testing systematically on

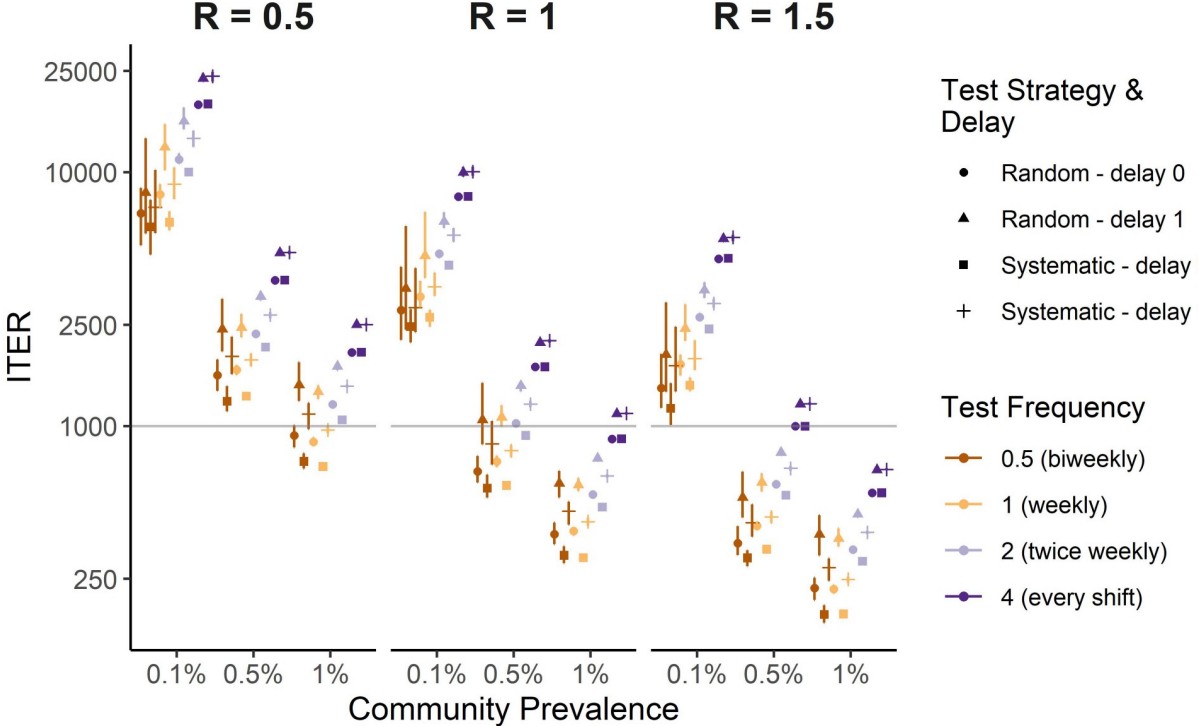

**Fig 4. Incremental test effectiveness ratio (ITER) from simulations across transmission scenarios and testing frequencies and strategies.** The ITER remains relatively low in higher transmission scenarios even at high ($f = 4$) testing frequencies, potentially favoring such high-frequency testing strategies when within-facility transmission ($R$) and/or community prevalence are high. The y-axis is log-transformed and the horizontal line at **ITER = 1000** is provided to aid visual comparison across scenarios. Error bars represent the interquartile range of expected infections derived from 100 simulations per scenario.

the first day of every other work week with no delay ($f = 0.5$, $d = 0$, Fig 4, see squares) leads to ITER = 180.89 (IQR 168.01–196.5), while increasing test frequency to weekly ($f = 1$) results in ITER = 181.72 (IQR 178.78–186.08), to twice weekly ($f = 2$): ITER = 293.02 (IQR 288.91–295.6), and to every shift ($f = 4$): ITER = 545.36 (IQR 541.58–550.61). These values approximately correspond to test positivity rates of 0.55%, 0.55%, 0.34%, and 0.18%. Fig 4 also provides an example reference line at *ITER* = 1000, corresponding to an approximate 0.1% test positivity, to demonstrate how testing frequency may be determined from the transmission scenario and target ITER, which may be influenced by the number of tests available.

Imperfect test sensitivity and self-isolation of infectious individuals were not found to meaningfully impact the main findings comparing testing strategies. Fig B in S1 File shows that test frequency and isolation delays are more influential on expected infections than test sensitivity, while Fig C in S1 File shows that self-isolation reduces the overall number of expected infections in simulations, but the superiority of systematic testing strategies persists even with perfect self-isolation of symptomatic infections.

## Discussion

This study builds on previous modeling and simulation analyses to demonstrate that systematic testing strategies with limited delays between test administration and isolation of infectious individuals can limit SARS-CoV-2 transmission. In particular, systematically testing workers early in their workweek prevents more infections than random testing strategies or those with a delay between testing and isolation. A major benefit of such strategies is that they do not require higher testing frequency, only a change in timing of when testing occurs. As such, there is substantial value in implementing systematic rapid testing at the beginning of the workweek in facilities at high risk for SARS-CoV-2.

The occurrence of pre- and asymptomatic SARS-CoV-2 transmission calls for systematic testing to be a key component of prevention strategies. Increasing the frequency of testing may be necessary in settings with high community prevalence or the opportunity for rapid spread within a facility (e.g. highly transmissible variants, low vaccination rates, inadequate mitigation practices). Lower thresholds than one expected infection event per ten days may also be necessary to prevent outbreaks in carceral facilities and other congregate settings. A prior analysis of publicly available CDCR case data estimated 46% of 118 SARS-CoV-2 introductions into resident populations from April 2020 to March 2021 across 35 facilities resulted in outbreaks of greater than 10 resident cases [32], though this estimate includes data from early in the pandemic when there were more fully susceptible individuals, fewer protocols to reduce transmission, limited testing resources, and lower vaccination coverage.

This study also utilized the ITER as a per-test measure of effectiveness for systematic testing across a range of frequencies and transmission scenarios. In resource-constrained environments in which tests are difficult to acquire (e.g., limited supply/funds), the ITER and its relationship to test positivity may be used to guide decisions on test frequency. The ITER may also be useful in situations where data on the cost per COVID-19 case and cost per test are available. In this case, the product of the ITER and the cost per test conducted provides the cost per case avoided due to the testing program. For facility management, any testing program that results in a lower cost per case avoided than cost per COVID-19 case would likely be deemed cost effective.

Even though systematic testing strategies reduce within-facility transmission, they are not capable of preventing all transmission events. Systematic testing represents one tool of many that could be implemented to prevent SARS-CoV-2 infections in congregate facilities. Facility-wide vaccination, universal masking, quarantine of individuals after exposure, avoiding

crowds, physical distancing, and proper ventilation all play an important role in mitigating SARS-CoV-2 transmission in congregate settings [33]. However, sometimes low vaccine acceptance rates among residents and staff in correctional settings coupled with more transmissible SARS-CoV-2 variants puts this population at continued risk of outbreaks. Implementing routine, systematic testing of staff for early identification of COVID-19 cases (including infections in vaccinated persons) is another layer of intervention that can help prevent outbreaks in congregate facilities.

## Limitations

Staff are not the only source of infection, as cases may also be imported via new resident intake, visitation, between-facility movement, and work programs where residents leave the facility during the day. In addition, the exclusion of notable COVID-19 prevention strategies (e.g. universal masking, physical distancing, proper ventilation, vaccination) and of additional testing due to symptoms or known contacts is a limitation. However, the sensitivity analysis implementing self-isolation suggests qualitative trends between testing strategies would persist, and only the magnitude of simulated infections would be reduced. Furthermore, these additional interventions would presumably act by reducing $\mathcal{R}$ and are therefore implicitly included in simulations with variable reproduction numbers.

The model also does not distinguish between staff-staff and staff-resident transmission within a simulated facility, but rather records the total number of infections expected assuming $\mathcal{R}$ remains constant. Estimation of staff-staff and staff-resident contact rates or reproduction numbers would enable more precise accounting and simulation of importation events and subsequent transmission within a facility. Given the goal of this analysis to investigate the utility of different testing strategies to prevent outbreaks, explicit simulation of transmission within the facility was deemed beyond the scope of the present analysis.

The community force of infection among staff is constant through time and across individuals. In reality, community prevalence can increase rapidly, necessitating a corresponding increase in test frequency. Heterogeneity among staff due to vaccination coverage, compliance with physical distancing and masking policies, household structure and/or health status, and other behavioral factors may also affect the rate of community spillover to staff and subsequently to facilities. Incorporating such variability in individual staff would likely increase the variance of results, but is not expected to appreciably affect key findings of the analysis.

Finally, the probability density function of the triangle distribution is used to represent SARS-CoV-2 viral dynamics and therefore infectiousness through time. Though this function captures the general viral dynamics profile seen previously [18, 21], other distributions or functions may also be applicable, though other analyses using more complex infectiousness profiles have yielded similar results [34].

## Future directions and conclusions

This modeling and simulation framework is applicable beyond COVID-19 in congregate settings. Other applicable settings may include the introduction of hospital acquired infections from newly admitted patients or from hospital staff [35], introduction of other respiratory pathogens such as influenza or pertussis into congregate settings [36], or tuberculosis transmission between communities and populations experiencing incarceration [37]. Accurate parameterization of key natural history traits of the pathogen in question such as the latent, incubation, and infectious periods is essential to estimate the impact of nonpharmaceutical interventions such as systematic screening testing [17]. Pathogens other than SARS-CoV-2

that cause symptoms prior to infectiousness ($t_{incubation} < t_{latent}$), for instance, may be more effectively controlled at lower cost via symptom screening and subsequent isolation [17].

In conclusion, these results suggest that aligning testing schedules with regular working schedules for staff in congregate settings, in addition to timely implementation of prevention strategies (e.g., isolation) can improve the efficacy of systematic screening testing. Two metrics, the number of expected infections within a facility and the ITER are presented to inform decisions on the frequency of systematic testing needed in different transmission scenarios to limit transmission under key thresholds. Based on these findings, congregate settings such as carceral facilities, nursing homes, schools, and more may be able to better prevent outbreaks through systematic testing of staff that is aligned with work schedules and is continued until community transmission or within-facility transmission potential are sufficiently reduced.

## Supporting information

**S1 File. Results of sensitivity analyses investigating test sensitivity and self-isolation.** (DOCX)

## Acknowledgments

We acknowledge the California Department of Corrections and Rehabilitation, California Correctional Health Care Services, and the CDC's COVID-19 Response for supporting this study.

## Author Contributions

**Conceptualization:** Christopher M. Hoover.

**Data curation:** Nicholas K. Skaff, Rena Fukunaga.

**Formal analysis:** Christopher M. Hoover, Nicholas K. Skaff.

**Funding acquisition:** Seth Blumberg.

**Investigation:** Christopher M. Hoover, Nicholas K. Skaff, Rena Fukunaga.

**Methodology:** Christopher M. Hoover, Nicholas K. Skaff, Seth Blumberg.

**Project administration:** Seth Blumberg, Rena Fukunaga.

**Resources:** Seth Blumberg, Rena Fukunaga.

**Software:** Christopher M. Hoover, Nicholas K. Skaff.

**Supervision:** Seth Blumberg, Rena Fukunaga.

**Validation:** Christopher M. Hoover.

**Visualization:** Christopher M. Hoover, Nicholas K. Skaff.

**Writing – original draft:** Christopher M. Hoover, Nicholas K. Skaff, Rena Fukunaga.

**Writing – review & editing:** Christopher M. Hoover, Nicholas K. Skaff, Seth Blumberg, Rena Fukunaga.

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
