## [Decision Letter · Decision Letter 0]

19 Oct 2022

PGPH-D-22-01197

Aligning staff schedules, testing, and isolation reduces the risk of COVID-19 outbreaks in carceral and other congregate settings: A simulation study

Dear Dr. Christopher Hoover

Thank you for submitting your manuscript to PLOS Global Public Health. After careful consideration, we feel that it has merit but does not fully meet PLOS Global Public Health’s publication criteria as it currently stands. Therefore, we invite you to submit a revised version of the manuscript that addresses the points raised during the review process.

We look forward to receiving your revised manuscript.

Kind regards,

Srinivasa Rao Mutheneni, PhD

Academic Editor

Journal Requirements:

1. We have noticed that you have uploaded Supporting Information files, but you have not included a list of legends. Please add a full list of legends for your Supporting Information files after the references list. 

2. In the online submission form, you indicated that your data will be submitted to a repository upon acceptance.  We strongly recommend all authors deposit their data before acceptance, as the process can be lengthy and hold up publication timelines. Please note that, though access restrictions are acceptable now, your entire data will need to be made freely accessible if your manuscript is accepted for publication. This policy applies to all data except where public deposition would breach compliance with the protocol approved by your research ethics board. If you are unable to adhere to our open data policy, please kindly revise your statement to explain your reasoning and we will seek the editor's input on an exemption. Please be assured that, once you have provided your new statement, the assessment of your exemption will not hold up the peer review process.

Additional Editor Comments (if provided):

Reviewers' comments:

Reviewer's Responses to Questions

**Comments to the Author**

1. Does this manuscript meet PLOS Global Public Health’s publication criteria? Is the manuscript technically sound, and do the data support the conclusions? The manuscript must describe methodologically and ethically rigorous research with conclusions that are appropriately drawn based on the data presented.

Reviewer #1: Yes

Reviewer #2: Yes

2. Has the statistical analysis been performed appropriately and rigorously?

Reviewer #1: Yes

Reviewer #2: Yes

3. Have the authors made all data underlying the findings in their manuscript fully available (please refer to the Data Availability Statement at the start of the manuscript PDF file)?

Reviewer #1: Yes

Reviewer #2: Yes

4. Is the manuscript presented in an intelligible fashion and written in standard English?

Reviewer #1: Yes

Reviewer #2: Yes

5. Review Comments to the Author

Reviewer #1: The authors have described the data inputs, and methods for this simulation study. The authors have also highlighted transparency and ethical aspects through deposition of data, and simulations in a public repository. The manuscript is acceptable in the current form.

Reviewer #2: This study examines the relationship between work schedules, testing schedules, and within-facility transmission of SARS-CoV-2 in carceral and other congregate settings using the data from California Department of Corrections and Rehabilitation (CDCR). I would like to congratulate the author(s) to this useful attempt, since limited attempts conducted. Though the manuscript is well-written and the subject addressed in this manuscript is worth of investigation, to allow for a better clarity for the reader and meaningful scientific findings the authors suggested to (a) explain methods section/analysis of data, source, assumptions and validation of data and results further, since methodology is hard to follow; (b) figures given in this manuscript are rather fuzzy and unclear, and give the figures with higher resolution; (c) Title could be modified to indicate the study objective instead of observed results and include study setting/country in the title; (d) extend further with possible interaction terms; (e) discuss the strengths and develop limitations further and especially for each limitation it would be good to say why you believe it is not a major or not affecting the results, including the sources of the data considered, along with implications for further development. Before publication, I would urge the authors to have another good 'go' at the manuscript, post-acceptance, as I am sure they will be able to make further minor improvements in structure and presentation.

In conclusion, though the authors have done a good job in addressing the topic, I feel, the manuscript is certainly less disjointed and needs some major improvements on its structure, methodology/analytical framework and presentation of observations etc., before possible consideration of publication for a better clarity and follow the reader.

6. PLOS authors have the option to publish the peer review history of their article (what does this mean?). If published, this will include your full peer review and any attached files.

**Do you want your identity to be public for this peer review?** For information about this choice, including consent withdrawal, please see our Privacy Policy.

Reviewer #1: **Yes: **Denny John

Reviewer #2: No

---

## [Decision Letter · Decision Letter 1]

8 Dec 2022

Aligning staff schedules, testing, and isolation reduces the risk of COVID-19 outbreaks in carceral and other congregate settings: A simulation study

PGPH-D-22-01197R1

Dear Christopher Hoover,

We are pleased to inform you that your manuscript 'Aligning staff schedules, testing, and isolation reduces the risk of COVID-19 outbreaks in carceral and other congregate settings: A simulation study' has been provisionally accepted for publication in PLOS Global Public Health.

Best regards,

Srinivasa Rao Mutheneni, PhD

Academic Editor

Reviewer Comments (if any, and for reference):

Reviewer's Responses to Questions

**Comments to the Author**

1. If the authors have adequately addressed your comments raised in a previous round of review and you feel that this manuscript is now acceptable for publication, you may indicate that here to bypass the “Comments to the Author” section, enter your conflict of interest statement in the “Confidential to Editor” section, and submit your "Accept" recommendation.

Reviewer #2: All comments have been addressed

Reviewer #3: All comments have been addressed

2. Does this manuscript meet PLOS Global Public Health’s publication criteria? Is the manuscript technically sound, and do the data support the conclusions? The manuscript must describe methodologically and ethically rigorous research with conclusions that are appropriately drawn based on the data presented.

Reviewer #2: Yes

Reviewer #3: Yes

3. Has the statistical analysis been performed appropriately and rigorously?

Reviewer #2: Yes

Reviewer #3: Yes

4. Have the authors made all data underlying the findings in their manuscript fully available (please refer to the Data Availability Statement at the start of the manuscript PDF file)?

Reviewer #2: Yes

Reviewer #3: Yes

5. Is the manuscript presented in an intelligible fashion and written in standard English?

Reviewer #2: Yes

Reviewer #3: Yes

6. Review Comments to the Author

Reviewer #2: I would like to thank the authors for addressing the minor concerns and providing the justifications adequately on reviewer concerns satisfactorily. The subject addressed in this manuscript is worth of investigation and recommend for possible consideration in the journal of PLOS Global Public Health.

Reviewer #3: Response to previous concerns is adequate.

7. PLOS authors have the option to publish the peer review history of their article (what does this mean?). If published, this will include your full peer review and any attached files.

**Do you want your identity to be public for this peer review?** For information about this choice, including consent withdrawal, please see our Privacy Policy.

Reviewer #2: No

Reviewer #3: No
